# Comprehensive Analyses of Coagulation Parameters in Patients with Vascular Anomalies

**DOI:** 10.3390/biom12121840

**Published:** 2022-12-08

**Authors:** Friedrich G. Kapp, Cedric Schneider, Annegret Holm, Hannah Glonnegger, Charlotte M. Niemeyer, Jochen Rößler, Barbara Zieger

**Affiliations:** 1Division of Pediatric Hematology and Oncology, Department of Pediatrics and Adolescent Medicine, Medical Center—University of Freiburg, Faculty of Medicine, University of Freiburg, 79106 Freiburg, Germany; 2VASCERN VASCA European Reference Centre, 75108 Paris, France; 3Vascular Biology Program, Department of Surgery, Boston Children’s Hospital, Harvard Medical School, Boston, MA 02115, USA; 4Division of Paediatric Hematology and Oncology, Department of Paediatrics, Inselspital, Bern University Hospital, University of Bern, 3010 Bern, Switzerland

**Keywords:** vascular anomalies, coagulopathy, localized intravascular coagulopathy, Kasabach–Merritt phenomenon, von Willebrand factor, platelet function

## Abstract

Background: Vascular anomalies comprise a diverse group of rare diseases with altered blood flow and are often associated with coagulation disorders. The most common example is a localized intravascular coagulopathy in venous malformations leading to elevated D-dimers. In severe cases, this may progress to a disseminated intravascular coagulopathy with subsequent consumption of fibrinogen and thrombocytes predisposing to serious bleeding. A separate coagulopathy is the Kasabach–Merritt phenomenon in kaposiform hemangioendothelioma characterized by platelet trapping leading to thrombocytopenia and eventually consumptive coagulopathy. Our previous work showed impaired von Willebrand factor and platelet aggregometry due to abnormal blood flow, i.e., in ventricular assist devices or extracorporeal membrane oxygenation. With altered blood flow also present in vascular anomalies, we hypothesized that, in particular, the von Willebrand factor parameters and the platelet function may be similarly impacted. Methods: We prospectively recruited 73 patients with different vascular anomaly entities and analyzed their coagulation parameters. Results: Acquired von Willebrand syndrome was observed in both of our patients with Kasabach–Merritt phenomenon. In six out of nine patients with complex lymphatic anomalies, both the vWF antigen and activity were upregulated. Platelet aggregometry was impaired in both patients with Kasabach–Merritt phenomenon and in seven out of eight patients with an arteriovenous malformation. Conclusions: The analysis of coagulation parameters in our patients with vascular anomalies advanced our understanding of the underlying pathophysiologies of the observed coagulopathies. This may lead to new treatment options for the, in part, life-threatening bleeding risks in these patients in the future.

## 1. Introduction

Vascular anomalies comprise a wide spectrum of rare disorders that are attributed to focal disruption in vascular development processes and are classified according to the International Society for the Study of Vascular Anomalies (ISSVA) [1]. Two distinct entities are differentiated: vascular tumors and vascular malformations. Vascular tumors are further subdivided into benign, locally aggressive, and malignant tumors. Vascular malformations can occur in each vessel type and are thus classified into capillary, venous, arteriovenous, and lymphatic malformations. Additionally, mixed malformations and malformations associated with other anomalies are also specified.

Patients with vascular anomalies may show impaired coagulation parameters that encompass two separate coagulopathies, which both can lead to consumption coagulopathy: (1) localized intravascular coagulopathy (LIC) observed in patients with slow-flow malformations such as venous and mixed lymphatic–venous malformations, and (2) Kasabach–Merritt phenomenon (KMP) in patients with kaposiform hemangioendothelioma (KHE) as well as Kasabach–Merritt-like phenomenon in patients with kaposiform lymphangiomatosis. LIC is often associated with pain and is caused by abnormal blood flow in the slow-flow malformation leading to thrombosis due to stasis and potential damage in the endothelium of malformed vessels [2]. The analysis of coagulation parameters in LIC show elevated D-dimers; patients with LIC and extensive disease may also show reduced fibrinogen and platelet counts indicating a transition to disseminated intravascular coagulopathy (DIC). In contrast, KMP is caused by the trapping of platelets in the vessels of the tumor [3]. This leads to an increase in the size of the tumor and a severe thrombocytopenia, which may be associated with the risk of relevant and, in part, life-threatening bleeding [4,5]. These two coagulopathies highlight the need for a hemostaseological workup in patients with vascular anomalies, especially in patients with extensive disease and with specific vascular tumors or malformations. While LIC and KMP are known complications of vascular anomalies, in-depth evaluation of coagulation parameters in patients with vascular anomalies are lacking. A thorough analysis of coagulation parameters may guide diagnostic and therapeutic approaches in patients with these rare diseases.

In this prospective, single-center study, we aimed to characterize coagulation and platelet parameters in patients with different vascular anomalies with a focus on the von Willebrand factor (vWF). We hypothesized that, with its origin in endothelial cells, the vWF may play a central role in coagulopathies in vascular anomalies. We showed in our previous work that vWF and platelet function are disturbed by abnormal blood flow as exemplified in patients with ventricular assist devices (VAD) or extracorporeal membrane oxygenation (ECMO) [6,7,8,9,10]. In these devices, the high molecular weight multimers of vWF are lost leading to a reduction in the vWF activity and ratio. Additionally, platelets are activated in VAD and ECMO by shear stress, which leads to a secondary exhaustion and platelet dysfunction. This can be alleviated by improving the flow characteristics of these assist devices [6]. This prompted us to include vWF parameters and platelet aggregometry into our analysis, hypothesizing that the pathomechanism of abnormal blood flow causing a coagulopathy in patients with external devices may also be observed in vascular anomalies.

## 2. Materials and Methods

This study involved human participants and was approved by the Ethics Committee of the University Hospital Freiburg (no. 434/16). Written informed consent to participate in this study was provided by the participant or the participants’ legal guardian. Patients were recruited prospectively from 2016 to 2020 in the outpatient clinic for vascular anomalies of the Department for Pediatric Hematology and Oncology of the University Hospital Freiburg. Mainly patients who had an indication for a blood analysis (e.g., due to pain, a suspected coagulopathy, or before imaging) were included in the study, thereby often selecting more severely affected patients. The vWF parameters were measured in each patient. In addition, we aimed to include basic hemostaseological parameters (e.g., platelet count, INR, PTT, D-dimers, fibrinogen) given that the quantity and quality of the material provided was sufficient. We also assessed platelet function, which was limited in some patients by the relatively large blood volume required for this assay (see Appendix A for details on the numbers of patients and tests). Results of previous and subsequent blood analyses that included parameters of interest for this study were also included in the data analysis.

### 2.1. Von Willebrand Factor Analyses

vWF antigen (vWF:Ag, normal 0.6–1.5 U/L), vWF:collagen binding activity (vWF:CB, normal 0.6–1.5 U/L), and vWF multimers were determined as previously described [8]. Briefly, vWF:Ag was measured in sodium citrate plasma using an in-house ELISA. Collagen type I (Nycomed Pharma, Unterschleissheim, Germany) was immobilized on a microtiter plate. vWF:CB in plasma was measured photometrically via ELISA technique. We calculated ratios of vWF:CB/vWF:Ag (normal ≥ 0.7). They reflect the biological activity of the available vWF to bind to collagen. vWF multimers were separated on sodium dodecyl sulfate-agarose gel and blotted on a PVDF (polyvinylidene fluoride) membrane to assess the HMW multimers. vWF was determined using appropriate primary and secondary antibodies (DAKO, Hamburg, Germany) and 3.30-diaminobenzidine/cobalt chloride (Bio-Rad, Munich, Germany). Standard human plasma (Siemens Healthcare Diagnostics) was used as control. Acquired von Willebrand syndrome (AVWS) was diagnosed if HMW multimers of vWF were missing and if the vWF:CB/vWF:Ag ratio was reduced. On the basis of visual inspection of the gel electrophoretic samples by a highly experienced examiner, the content of vWF multimers was classified as “normal,” “borderline” (reduced but not entirely absent), and “pathological” (severely reduced).

### 2.2. Platelet Aggregometry

Blood samples were obtained from the patients and platelet rich plasma (PRP) was prepared from citrated blood. Platelet agglutination/aggregation was analyzed on platelet aggregometer APACT 4 (Haemochrom Diagnostica, Essen, Germany) using following agonists: adenosine diphosphate (ADP; 4.0 and 10 μmol/L; Sigma Aldrich, St. Louis, MO, USA), arachidonic acid (0.3 mg/mL; möLab, Langenfeld, Germany), collagen (2.0 μg/mL; TAKEDA, Austria), epinephrine (8 μmol/L; Aventis, BRD), and ristocetin (1.2 mg/mL; American Biochemical and Pharmaceutical Ltd., Epsom, UK) [11].

### 2.3. Platelet Flow Cytometry

The flow cytometric assessment of platelets was performed using FACSCalibur (Becton Dickinson, Heidelberg, Germany) [12]. Diluted PRP aliquots (5 × 10^7^/mL) were fixed and stained with FITC-labeled monoclonal surface antibody against glycoproteins (GP) CD41 (GPIIb/IIIa-complex, integrin αIIbβ3), CD42a (GPIX) and CD42b (GPIb) (Coulter, Immunotech, Marseille, France). FITC-labeled anti-vWF (Bio-Rad AbD Serorech, Puchheim, Germany) and Alexa Fluor 488-labeled anti-fibrinogen (Invitrogen, Waltham, MA USA) were used to stain the platelets. In the presence of 1.25 mM Gly-Pro-Arg-Pro (Bachem, Bubendorf, Switzerland) diluted PRP (5 × 10^7^ platelets/mL) was stimulated with different concentrations of thrombin (0, 0.05, 0.1, 0.2, 0.5, and 1 U/mL; Siemens Healthineers, Marburg, Germany) to conduct the CD62 and CD63 expression analyses. Additionally, the platelets were stained with monoclonal FITC-labeled anti-CD62 (P-selectin) and anti-CD63 antibodies (lysosomal membrane associated glycoprotein 3, Immunotech, Marseille, France). Data of patients and controls (day control and 20 independent measurements from 10 controls as mean ± standard error of the mean, SEM) were analyzed using GraphPad Prism software (version 8, San Diego, CA, USA).

## 3. Results

### 3.1. Patient Characteristics

Seventy-three patients were included in this study. In total, 57.5% of the patients in the cohort were male. The median age at the time of study participation was 14 years (range 0.1–67 years). Blood for coagulation studies was drawn using routine blood tests. A total of 158 blood samples were analyzed. Thirty of the 73 patients (41.1%) enrolled were diagnosed with a venous malformation (VM). This reflects first, that this disease entity is the most common one that we treat in our specialized center for vascular anomalies and, second, that patients with VM most often require a blood analysis, e.g., due to pain and a suspected coagulopathy. Nine patients (12.3%) were diagnosed with a complex lymphatic anomaly (CLA), including three patients with Gorham–Stout disease (GSD) and six patients with a generalized lymphatic anomaly (GLA).

In 11/73 patients (15.1%) arteriovenous malformations (AVMs) were present, including 1 patient with a pulmonary AVM due to a hereditary hemorrhagic telangiectasia, 1 patient with an AV fistula on the basis of a CLOVES syndrome, and 2 patients with CM-AVM with AV malformations. Three patients (4.1%) had a capillary malformation (CM). A “mixed” malformation was present in 16 patients (21.9%); this category comprised 8 patients with mixed lymphatic–venous malformations and 8 patients with PIK3CA-related overgrowth spectrum (PROS) disease.

Four of the 73 patients (5.5%) were diagnosed with vascular tumors, including 2 patients with KHE, 1 patient with multiple infantile hemangiomas and aplasia of the vena cava, and 1 patient with an epithelioid hemangioma (Table 1 and Appendix A). All patients included had at least one analysis of vWF parameters during the course of this study (Appendix A for details).

### 3.2. Plasmatic Coagulation Parameters

The INR was increased above 1.2 in 7/70 patients (10.0%); 4 of these 7 patients had large VM (13.3% of VM patients), 1 patient suffered from a CLA (11.1% of CLA patients), and 2 patients had a KHE (Table 2).

Elevated D-dimers were common in our cohort with D-dimers frequently >2 mg/L FEU: 12/29 patients with VM (41.4% of VM patients in our cohort with available D-dimer analysis), 6 patients with CLA (66.7% of CLA patients), 4/11 patients with “mixed” (36.4% of patients with mixed malformations), and in both patients with KHE while suffering from KMP (Table 2).

Reduced fibrinogen (<130 mg/dL) was observed in 3/29 patients with a VM (10.3% of VM patients). In two of these three VM patients (patient 19 and 30), the platelet counts were reduced and the INR and aPTT were elevated, indicative of a DIC. The fibrinogen was also reduced in both patients with a KHE during KMP (Table 2).

### 3.3. Von Willebrand Factor Parameters

Most patients of our cohort showed normal vWF parameters (Figure 1).

Both patients with KHE developed large hematomas, which prompted further diagnostic workup. We observed AVWS in both patients with a KHE during KMP: both patients showed pathological vWF multimers (loss of high molecular weight multimers), a reduced vWF:CB/vWF:Ag ratio (0.65 and 0.47, respectively), and a reduced factor VIII activity (37% and 64%, respectively). One patient had a reduced vWF:Ag (nadir at 0.35), KMP led to thrombocytopenia (17 and 22 G/L, respectively), and DIC with elevated D-dimers and reduced fibrinogen in both patients (Appendix A). However, patient 70 with KHE also showed the loss of high molecular multimers at two different timepoints: one while D-dimers and fibrinogen were normal, and one while Quick (INR) and PTT were normal. One patient received multiple transfusions with fresh-frozen plasma. A whole-body MRI was performed due to unexplained DIC, which then led to the diagnosis of KHE in both patients.

In addition to the two patients with KMP and AVWS, the vWF:CB/vWF:Ag ratio was reduced below 0.7 in three other patients: one patient with a VM and DIC (platelet count 146 G/L, fibrinogen 62 mg/dL) also had borderline/slightly reduced vWF multimers. One patient with a large AVM of the left shoulder and arm also had a slight loss of high molecular weight multimers. One patient had a CLA and strongly elevated D-dimers (19.6 mg/L FEU) (Table 3 and Appendix A).

This did not occur in a subset of patients with CLA, in which six out of nine CLA patients (66.7%) even showed an increase in vWF:Ag and vWF:CB (Table 3). Of these six CLA patients with an upregulated vWF:Ag and vWF:CB (with a normal ratio and only slightly elevated D-dimers), five patients showed an increased factor VIII activity, indicating endothelial activation (Table 4). The remaining patient with a normal factor VIII activity had a reduced vWF:CB/vWF:Ag ratio. Notably, this was the patient with strongly elevated D-dimers of 19.6 mg/L FEU and a fibrinogen of only 147 mg/dL.

### 3.4. Platelet Aggregometry

Both patients with KHE during KMP had impaired platelet function and showed severely impaired platelet aggregation at different time points of the disease course, indicating a relevant and persisting thrombocytopathy even after the improvement of thrombocytopenia in one patient.

Seven out of eight patients (87.5%) with an arteriovenous malformation (AVM) and available thrombocyte aggregation showed an impaired platelet activation after stimulation with 4 µmol/L ADP. Five of these patients (representing 62.5% of AVM patients) also showed an impaired platelet activation after stimulation with 8 µmol/L epinephrin. Four out of these five also had a thrombocytopathy after stimulation with 10 µmol/L ADP (Table 5).

In addition, we performed flow cytometric analyses in four AVM patients: patient 41 (PWS), patient 45 (PWS), patient 49 (AVM of the shoulder), and patient 50 (AV fistula (wrist) and previous AVM of the spine in CLOVES syndrome) observed a decreased expression of CD63 hinting at an impaired delta granule secretion in three out of the four patients. CD62 was borderline in one patient. vWF-binding after activation with ristocetin was borderline in three out of four patients. Fibrinogen-binding after ADP-activation was pathologic in one patient and borderline in one patient (Appendix A; more detailed FACS data are available in Appendix A).

## 4. Discussion

This study is among the first to perform a comprehensive analysis of coagulation parameters in patients with vascular anomalies. In addition, we studied platelet function using platelet aggregometry. We focused on the analysis of vWF and platelet aggregometry based on our hypothesis that the abnormal blood flow within the lesions may lead to an alteration of these parameters.

Regarding the global coagulation parameters (INR, aPTT, fibrinogen, D-dimers), we observed the previously described hemostaseological complications of patients with vascular anomalies, especially DIC in extensive VMs and KMP in KHE.

D-dimers were frequently elevated: pronounced elevation was seen in VMs, mixed malformations and CLAs, as well as in patients with KHE. This high frequency of hemostaseological abnormalities likely reflects a selection bias in our patient cohort, as these patients were treated in a highly specialized center for vascular anomalies with a higher number of severely affected patients. Moreover, we only included patients that had a clinical indication for a blood analysis, thereby further selecting patients with a more severe disease condition at the time of the blood drawing. This selection bias should be considered and makes the results of this study only applicable to severely affected patients.

Patients with KHE showed the known changes in coagulation parameters during KMP [3]. Intriguingly, both of our patients with KMP based on KHE developed AVWS. One of these patients also showed reduced vWF multimers, while fibrinogen and D-dimers were within the normal range. AVWS may be a novel characteristic of KMP and allows for further pathophysiological insights into this rare disease; however, it might be difficult to discriminate between this and a general loss of coagulation factors in late stage DIC. Furthermore, both patients showed abnormal platelet function, which was likely due to a primary activation of platelets by increased shear stress within the tumor endothelium and secondary exhaustion of the platelets.

Therapeutically, some clinical centers use ASS to reduce platelet trapping within the tumor vasculature [13]. Considering the already abnormal platelet aggregation, the risks and benefits of this approach would need to be cautiously evaluated in the future. In both of our KHE patients, the diagnosis was made by whole-body MRI, highlighting the crucial role of imaging in patients with unexplained coagulopathies. Based on this experience, we would recommend that in patients presenting with increased D-dimers, reduced platelet counts, and an elevated INR without an increased C-reactive protein or other signs of infection, KHE should be considered as a differential diagnosis. A whole-body MRI is warranted in these cases and may lead to the diagnosis as it did in our two patients with KHE presented in this study.

In patients with vascular anomaly entities other than KHE, vWF:Ag and vWF:CB were within normal ranges in most patients.

Intriguingly, patients with CLAs showed a marked elevation of vWF:Ag and vWF:CB—a finding that was unexpected and has not been described for these entities before. In five out of six CLA patients with an elevation of vWF:Ag (and in five out of five with an elevated vWF:CB), factor VIII was also elevated, indicating that vWF elevation also led to a higher than normal stabilization of factor VIII. We hypothesize that this elevation of vWF may be due to an endothelial activation. Whether this is due to activation caused by the malformed lymphatic endothelium or due to inflammation that often is associated with CLAs, requires further investigation.

Of interest, seven out of eight patients with AVMs also showed impaired platelet aggregation, indicating a primary activation and secondary exhaustion of platelets due to increased shear stress associated with the characteristic high-flow conditions of AVMs (Figure 2). This finding could also be corroborated by a decreased expression of CD63, hinting at an impaired delta granule secretion of platelets as assessed by flow cytometry in four AVM patients. Platelet aggregation is critical for primary hemostasis, especially in mucosal areas. Hence, the impaired platelet function in these patients may explain a higher tendency of mucosal bleeding in patients specifically with facial/mucosal AVMs. Intriguingly, the finding of impaired platelet function in patients with AVMs was reminiscent of the coagulopathy found in patients with VAD or ECMO. However, the severity of the coagulopathy in AVMs was less pronounced, which may indicate that the blood flow in these lesions is still more physiological than the one in assist devices or ECMO.

## 5. Conclusions

We present results of a comprehensive, prospective, and single-center analysis of coagulation parameters with a focus on the vWF and platelet function parameters that provide important and novel insights into the pathophysiology of vascular anomalies. We show that CLAs are associated with increased vWF parameters and elevated factor VIII activity. Patients with KHE may develop impaired platelet function and AVWS during KMP. vWF parameters are severely affected in patients with KMP and vWF activity is reduced, similar to what has been observed in patients with VAD or ECMO. The fast-flow conditions in AVMs may trigger platelet activation that can ultimately lead to platelet exhaustion, which is similar but not as pronounced as in patients with VAD or ECMO. There is a selection bias towards more severely affected patients due to, first, more severely affected patients being treated at our highly specialized VA center and, second, a blood drawing being necessary more often in these patients.

Understanding the hemostaseological consequences in different vascular anomaly entities may enable us to make patient management safer and contribute to the development of medical treatment options that alleviate bleeding risks in these patients in the future.

## Figures and Tables

**Figure 1 biomolecules-12-01840-f001:**
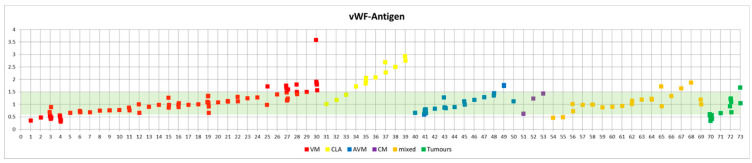
Graphical representation of the vWF:Ag in patients with different entities of vascular anomalies. The reference range of vWF:Ag is highlighted in green.

**Figure 2 biomolecules-12-01840-f002:**
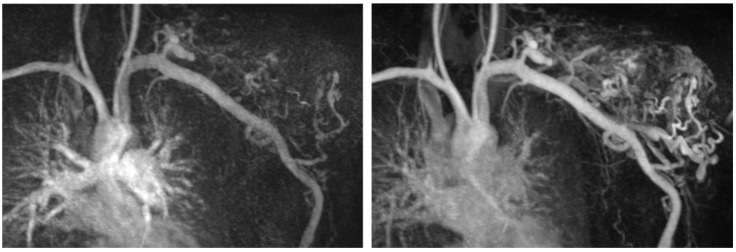
MRI TWIST-angiography of a patient with an AVM of the left shoulder highlighting the fast-flow conditions within the AVM.

**Table 1 biomolecules-12-01840-t001:** Patient characteristics. Abbreviations: VM = venous malformation, CLA = complex lymphatic anomaly; AVM = arteriovenous malformation, CM = capillary malformation.

Patient Characteristics		
Sex:	n=	%
Male	42	57.5
Female	31	42.5
Blood draws	158	
Age at blood draws:	Median	Range
(years)	14	0.1–67
Diagnosis:	n=	%
VM	30	41.1
CLA	9	12.3
AVM	11	15.1
CM	3	4.1
Mixed	16	21.9
Tumor	4	5.5

**Table 2 biomolecules-12-01840-t002:** Alterations in parameters of plasmatic coagulation specified by disease. Additionally, PTT was prolonged (>40 s) in 3 patients (2× VM, 1× tumor/KHE); thrombocytes were decreased in 4 patients (2 VM, 2 tumor/KHE).

Plasmatic Coagulation											
Patients with INR > 1.2	n=	of n=	%	Patients with D-dimers > 2 mg/L FEU	n=	of n=	%	Patients with fibrinogen < 130 mg/dL	n=	of n=	%
VM	4	30	13.3	VM	12	29	41.4	VM	3	29	10.3
CLA	1	9	11.1	CLA	6	9	66.7	CLA	0	8	0.0
AVM	0	11	0.0	AVM	0	10	0.0	AVM	0	11	0.0
CM	0	2	0.0	CM	0	1	0.0	CM	0	2	0.0
Mixed	0	14	0.0	Mixed	4	11	36.4	Mixed	0	12	0.0
Tumor	2	4	50.0	Tumor	2	3	66.7	Tumor	2	4	50.0

**Table 3 biomolecules-12-01840-t003:** Alterations in vWF:Ag specified by disease. vWF:CB was changed in the same patients.

vWF:Ag							
vWF-Ag < 0.6	n=	of n=	%	vWF-Ag > 1.5	n=	of n=	%
VM	4	30	13.3	VM	4	30	13.3
CLA	0	9	0.0	CLA	6	9	66.7
AVM	0	11	0.0	AVM	1	11	9.1
CM	0	3	0.0	CM	0	3	0.0
Mixed	2	16	12.5	Mixed	3	16	18.8
Tumor	1	4	25.0	Tumor	1	4	25.0

**Table 4 biomolecules-12-01840-t004:** Details von vWF parameters in patients with CLA. Bold font was used to label abnormal values.

ID	Disease	vWF:Ag	vWF:CB	vWF-CB/Ag	vWF-Multimers	F VIII
31	GSD	1.02	0.83	0.81	normal	
32	GSD	1.18	1.49	1.26	normal	110
33	GLA	1.39	1.28	0.92	normal	140
34	GLA	**1.72**	**2.81**	1.63	normal	**170**
35	GLA	**1.84**	**1.93**	1.05	normal	150
		**2.07**	**2.71**	1.31	normal	**160**
		**1.96**	**2.11**	1.08	normal	
36	GLA	**2.09**	**2.03**	0.97	normal	**188**
37	GSD	**2.69**	**1.87**	**0.69**	normal	120
		**2.29**	**1.62**	0.71	normal	
38	GLA	**2.51**	**2.04**	0.81	normal	**190**
39	GLA	**2.93**	**2.56**	0.87		**283**
		**2.76**	**2.41**	0.88	normal	

**Table 5 biomolecules-12-01840-t005:** Alterations in thrombocyte aggregation specified by disease.

Platelet Aggregometry											
Patients with <70% aggregation after stimulation with 4 µmol/L ADP	n=	of analyzed n=	%	Patients with aggregation <70% after stimulation with 8 µM Epinephrin	n=	of analyzed n =	%	Patients with aggregation <70% after stimulation with 10 µmol/L ADP	n =	of analyzed n=	%
VM	19	24	79.2	VM	11	24	45.8	VM	6	16	37.5
CLA	5	7	71.4	CLA	1	7	14.3	CLA	0	5	0.0
AVM	7	8	87.5	AVM	5	8	62.5	AVM	4	7	57,1
CM	2	3	66.7	CM	0	3	0.0	CM	0	0	---
Mixed	7	11	63.6	Mixed	2	11	18.2	Mixed	0	5	0.0
Tumor	3	4	75.0	Tumor	2	4	50.0	Tumor	1	3	33.3

## Data Availability

The data supporting the reported results can be found in the full table in the Appendix A.

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
