# Peer review of "Comprehensive Analyses of Coagulation Parameters in Patients with Vascular Anomalies"

_biomolecules, 2022, doi:10.3390/biom12121840_

Round 1

Reviewer 1 Report

1. The authors reported a study on the coagulation parameters measured in a collection of 73 patients with various types of vascular malformations.  This is an interesting study but the study design, patient selection bias, and the heterogeneity of the patients make interpretation and generalization of the results difficult.

2. The Abstract.  The Methods section is too brief.  In particular, does “different vascular anomaly entities” imply all kinds of vascular anomalies?  If not, please elaborate what anomalies are included or excluded.  Please also provide an outline of the coagulation parameters prospectively allocated for the investigative process.  In the Conclusion, please comment if the hypothesis that the coagulation abnormalities found in this group of patients are similar to patients on assisted ventricular device or extracorporeal membrane oxygenation is supported or not.

3. The Introduction.  The authors mentioned in their prior works that von Willebrand factor and platelet function were disturbed by abnormal blood flow.  In what ways were von Willebrand factor and platelet function abnormal?  Please explain so that the readers can compare directly with the findings from the present study.

4. Materials and Methods.  Please elaborate clearly on the recruitment or study period, inclusion and exclusion criteria.  If this was a prospective study with ethical committee approval, why not all patients underwent the same, complete list of investigations?  For patients who might have undergone interventional treatment, were they investigated prior to such interventions?  If not, how did the authors account for the laboratory findings were induced by the vascular anomalies and not from the interventions?  For platelet aggregometry, was there a minimal platelet count requirement?  If platelet aggregation tests were done in patients with significant thrombocytopenia, how did the authors conclude their findings?

5. The way patients are referred/recruited into the study and the clinical setting in which they are subjected to the coagulation studies are important to define the limitations of the study and if the findings can be extrapolated into the real-world patients.  Such limitations should be properly addressed at the end of the Discussion or included in the Conclusion.

6. It appears obvious to me that the reduction in von Willebrand factor as detected in this group of patient, either with a loss of the von Willebrand factor multimers or a reduction in the vWF:CB/vWF:Ag ratio is just a part of the consumptive coagulopathy or disseminated intravascular coagulation.  This is perhaps not an unexpected finding.  Should the authors have measured other coagulation factors individually, I would not be surprised to see other kinds of “acquired coagulopathies”.  To name such conditions as acquired von Willebrand syndrome or disease is not appropriate.

7. In Conclusion, the statement that in patients with an unclear coagulopathy, an MRI may lead to the diagnosis of a KHE is not related to the design and findings from the current study and hence should not be included.

Reviewer 2 Report

Reviewer - biomolecules-2044384

Dear Authors

Excellent study on the risk of AVWS development among vascular anomaly through KMP, I congratulate.

But I feel you ought to spare a few paragraphs to introduce the difference between ‘vascular tumor’ and ‘vascular malformation’ more clearly since both vascular disorders were grouped together under one group of ‘vascular anomaly’, Indeed, it would help ordinary readers to understand better such potential value of your discovery/hypothesis for the AVWS for its clinical implication to this unique group of coagulopathies.

All the best,

A Reviewer

Reviewer 3 Report

The authors' main aim is to study whether Von Willebrand factor activity and platelet aggregometry are affected in 73 patients with six different types of vascular anomaly. They observed acquired von Willebrand syndrome and impaired platelet aggregation in two patients with KMP. They also observed impaired platelet aggregation in patients with an AVM.

Major comments:

1. In the introduction section, the authors mentioned that they analyzed coagulation parameters and platelet parameters with a focus on vWF. This information is missing in the abstract. Please add in the abstract that the main aim of this study is to analyze Von Willebrand factor activity and platelet aggregometry among the patients.  

2. Please, describe the abbreviations shown in table 1 in the table 1 legend.

3. In the result section, the authors first described the D-dimer levels and then the INR, but in Table 2, the INR is presented first. The reviewer suggests switching the order that these parameters appear in the table for consistency within the text.

4. Please, include in table 2 the increased aPTT and decreased platelets count values among patients.

5. The authors reported that 7/8 patients with AVM have impaired platelet aggregation, but in the abstract, the authors reported that 8/9 have impaired platelet aggregation. Please, clarify this discrepancy.

6. Table 5 only shows platelet aggregation with 8 µM Epinephrin. Please, include the results obtained with ADP (4 uM and 10 µM) collagen and ristocetin in Table 5?

7. On page 4, sentence 190, the authors talked about Table 4, but the data about platelet aggregation is in Table 5. Please, fix this error. 

8. Please, show the raw data for the flow cytometry experiment compared to the control group in Supplement table 3.
